# Robust Auction Design in the Auto-bidding World

**Santiago Balseiro**
Columbia University and Google
srb2155@columbia.edu

**Yuan Deng**
Google
dengyuan@google.com

**Jieming Mao**
Google
maojm@google.com

**Vahab Mirrokni**
Google
mirrokni@google.com

**Song Zuo**
Google
szuo@google.com

## Abstract

In classic auction theory, reserve prices are known to be effective for improving revenue for the auctioneer against quasi-linear utility maximizing bidders. The introduction of reserve prices, however, usually do not help improve total welfare of the auctioneer and the bidders. In this paper, we focus on value maximizing bidders with return on spend constraints—a paradigm that has drawn considerable attention recently as more advertisers adopt auto-bidding algorithms in advertising platforms—and show that the introduction of reserve prices has a novel impact on the market. Namely, by choosing reserve prices appropriately the auctioneer can improve not only the total revenue but also the total welfare. Our results also demonstrate that reserve prices are robust to bidder types, i.e., reserve prices work well for different bidder types, such as value maximizers and utility maximizers, without using bidder type information. We generalize these results for a variety of auction mechanisms such as VCG, GSP, and first-price auctions. Moreover, we show how to combine these results with additive boosts to improve the welfare of the outcomes of the auction further. Finally, we complement our theoretical observations with an empirical study confirming the effectiveness of these ideas using data from online advertising auctions.

## 1 Introduction

As auto-bidding—the practice of using optimization algorithms to procure advertising slots—is becoming the prevalent option in online advertising, a growing body of work is revisiting auction theory from the lens of the auto-bidding world [Aggarwal et al., 2019, Babaioff et al., 2021, Balseiro et al., 2021a, Deng et al., 2021b, Golrezaei et al., 2021c]. A benefit of auto-bidding is that it simplifies advertisers' bidding process by asking advertisers for their high-level goals and then bidding on behalf of the advertisers.[1] The main difference with classic auction theory stems from the model adopted for agent behavior. Unlike the classic *utility maximization* model, where each agent aims to maximize its own quasi-linear utility given by the difference between value and payment, the behavior of an auto-bidding agent is determined by its underlying optimization algorithm. In particular, the prevalent adopted model for the behavior of auto-bidding agents is that of *value maximization*, where each agent aims to maximize total values subject to an *ROS (return on spend) constraint* on the average spend per opportunity. For example, two common auto-bidding strategies are target CPA (cost per acquisition) and target ROAS (return on ad spend) auto-bidding, in which an algorithm optimizes the

---

[1]See https://support.google.com/google-ads/answer/6268637 and https://www.facebook.com/business/m/one-sheeters/value-optimization-with-roas-bidding; and Aggarwal et al. [2019], Balseiro et al. [2021a], Deng et al. [2021b] for more background introductions.

35th Conference on Neural Information Processing Systems (NeurIPS 2021).

total *value* (e.g., the number of conversions) subject to an ROS constraint specified by the advertiser (e.g., the average spend for each conversion should not exceed a pre-specified target).

One surprising observation when agents are value maximizers is that the Vickrey-Clarke-Groves (VCG) auction, which is truthful and efficient for utility maximizers, is no longer truthful nor efficient. Value maximizers have incentives to strategize their bids and, moreover, there exist auction instances where the social welfare at equilibria under the VCG auction is only $1/2$ of the optimal social welfare [Aggarwal et al., 2019, Deng et al., 2021b]. This is because the ratio between bids and values can be very different for each value-maximizing agent, and an agent with a low value may end up outbidding other bidders with high values in some auctions, creating allocation inefficiency. While there have been recent work addressing this problem, most previous studies have several shortcomings, e.g., they reduce such efficiency loss by introducing boosts that explicitly or implicitly rely on the accurate knowledge of the advertiser values [Deng et al., 2021b], or they aim to optimize for either value-maximizing buyers or utility maximizing buyers (and not a mixture of auto-bidders) [Balseiro et al., 2021a], or they aim to optimize only welfare and not revenue [Deng et al., 2021b].

In this paper, we aim to address the above shortcomings and propose simple auctions taking inaccurate value signals as additional inputs and we prove that the auctions have robust approximation guarantees in terms of both social welfare and revenue. Such inaccurate value signals could be the outcomes of some machine learning models that have bounded multiplicative errors on advertiser values.[2] Our key theoretical result is a generic lemma that transfers the accuracy of the value signals into approximation guarantees on both welfare and revenue for various auctions. Moreover, our main theorems not only work in the pure auto-bidding world, but also in mixed environments in which both value maximizers and utility maximizers coexist. Furthermore, the approximation guarantees apply to a very general set of market outcomes, i.e., they hold as long as no bidder is using a bidding profile that is always dominated under any competing bids. Such a broad solution concept includes classical notions such as complete-information Nash equilibrium as refinements and, as we discuss in our empirical results, additionally provides guarantees that hold along the convergence path when agents individually optimize their strategies using simple update rules.

Our auctions are based on VCG or generalized second-price auctions (GSP) and use the inaccurate signals as reserve prices and/or as (additive) boosts. Reserve prices are minimum prices the agents should clear to win the auction. Boosts are additive transformations on top of bids, i.e., agents are ranked based on the sum of their bids and boosts. Moreover, boosts are subtracted from standard payments to maintain incentive compatibility for utility maximizers. Intuitively, adding properly chosen boosts can push the auction to a more efficient outcome [Deng et al., 2021b]. On the other hand, in contrast to the classic utility maximization setup in which reserve prices usually do not help improve social welfare, modest reserve prices in fact may improve social welfare when there are value-maximizing agents in the market. The key difference is that in truthful auctions like the VCG auction, utility-maximizers never react to reserve prices, but value-maximizers do react to reserve prices to satisfy their ROS constraints. It turns out that, with properly chosen reserve prices, one can eliminate *bad* bidding strategies by reducing the set of bidding strategies that could be best responses to other bidders' strategies. As a result, the remaining *good* bidding strategies lead to an outcome with better social welfare guarantees.

## 1.1 Our results

We propose a set of simple auctions and prove approximation guarantees both in terms of social welfare and revenue. In particular, these approximation guarantees for the proposed auctions are all tight except for the social welfare approximation for GSP with reserve (Appendix E). The conclusions are *robust* against: i) signal inaccuracy, ii) agent behavior models (value/utility maximization or intermediate), iii) system status beyond equilibrium. The following Table 1 summarizes most of our approximation results, where $\gamma \in [0, 1]$ describes the approximation accuracy of the signals and $\lambda \in [0, 1]$ indicates a hybrid behavior model between pure value maximization and pure utility maximization (e.g., $\lambda = 0$ for pure value maximization and $\lambda = 1$ for pure utility maximization). All formal definitions will be given later in Section 2.

---

[2]The learning task of the signals can be a complex problem given the potential interaction with advertiser incentives. In this paper, we are agnostic about how the signals are learned and hence the learning problem is out of the scope. Nevertheless, the line of work on incentive-aware learning [Epasto et al., 2018, Golrezaei et al., 2021a] could be relevant to avoid or mitigate inappropriate incentives.

Table 1: Social welfare and revenue approximation guarantees for different auctions with $\gamma$-approximate value signals ($\gamma \in [0, 1]$). For boosts we need: signal $\in [\gamma \cdot \nu \cdot \mathsf{value}, \nu \cdot \mathsf{value})$ with $\nu \geq 1$; for reserves we need: signal $\in [\gamma \cdot \mathsf{value}, \mathsf{value})$.

| Auction | Behavior model[1] | Social welfare[2] | Revenue | Theorem |
|---|---|---|---|---|
| VCG with reserve | $\lambda \in [0, 1]$ | $1/(2 - \gamma)$ | $\gamma$ | Corollary 4.2 |
| VCG with boost | $\lambda \in [0, 1]$ | $1/(2 - \gamma)$ | - | Corollary 4.3 |
| VCG with reserve and boost | $\lambda \in [0, 1]$ | $(1 + \gamma)/2$ | $\gamma$ | Corollary 4.4 |
| GSP with reserve | $\lambda \in [0, 1]$ | $\gamma$ | $\gamma$ | Corollary 4.8 |
| GSP with reserve and boost | $\lambda = 0$ | $(1 + \gamma)/2$ | $\gamma$ | Corollary 4.6 |

[1] $\lambda = 0$: value maximization; $\lambda = 1$: utility maximization; $\lambda \in (0, 1)$: intermediate models (see Program (1)).
[2] For $0 < \gamma < 1$, $(1 + \gamma)/2 > 1/(2 - \gamma) > \gamma$.

Observe that in a world with value maximizers only (i.e., $\lambda = 0$), if all the value maximizers can hit their targets (see Program (1)), the revenue equals to the social welfare. In such an environment, our social welfare guarantees can directly imply the same revenue guarantees under the condition that value maximizers all hit their targets.

We provide a general framework for proving approximation results in the presence of value maximizers through a novel technical lemma (Lemma 3.1). In fact, our approximation results are mostly derived from this lemma. Finally, we conduct empirical analyses with semi-synthetic data and validate our theoretical findings for the performance of our mechanisms in VCG auctions.

## 1.2 Related Work

As a central topic in economic study, since the seminal work of Vickrey–Clarke–Groves (VCG) auctions [Clarke, 1971, Groves, 1973, Vickrey, 1961] and Myerson's auction [Myerson, 1981], auction design has been successfully deployed in many different fields. Examples includes combinatorial auctions for reallocating radio frequencies [Cramton et al., 2006] and generalized second-price auctions as well as dynamic auctions for online advertising [Aggarwal et al., 2006, Edelman et al., 2007, Mirrokni et al., 2020, Varian, 2007].

In contrast to works assuming utility-maximizing agents, a growing body of work has recently focused on auto-bidders such as target CPA bidders and target ROAS bidders. Aggarwal et al. [2019] find the optimal bidding strategies for a general class of auto-bidding objectives and prove the existence of pure-strategy equilibrium. Deng et al. [2021b] show how boosts can be used to improve the efficiency guarantees when target CPA and target ROAS auto-bidders coexist. Balseiro et al. [2021a] characterize the revenue-optimal single-stage auctions with either value-maximizers or utility-maximizers with ROS constraints under various information structure. Golrezaei et al. [2021c] study the auction design for utility-maximizers with ROI constraints, where the behavior model is equivalent to the special case of ours with $\lambda = 1/(1 + \text{minimal ROI})$. Besides the model of constrained optimization, there is another generalization of the utility models with ROI constraints adopted by Babaioff et al. [2021], Goel et al. [2014, 2019]. In particular, the negative payment term in the utility function is replaced by a general cost, which is often a convex function of the payment.

Independent of the above, the term of value-maximizer has been used in some other works but with quite different mathematical models Fadaei and Bichler [2016], Wilkens et al. [2016, 2017]. The main distinction is that in these models, the target ROS constraints are imposed on a single auction rather than across a set of auctions, hence the bidders become insensitive to the marginal tradeoff between values and payments, leading to different behavior patterns.

Prior to these works, the simplest auto-bidding model is budget optimization with utility-maximizers [Borgs et al., 2007]. Pai and Vohra [2014] characterize the revenue-optimal auction of utility-maximizing bidders with budget constraints. For applications in online advertising, Balseiro and Gur [2019] develop budget management strategies that are no-regret in a long run, which is extended in more complex settings [Avadhanula et al., 2021, Balseiro et al., 2020b, 2021b, Celli et al., 2021]. Balseiro et al. [2017, 2020a] provide a thorough study to compare different commonly used budget management strategies in practice. Conitzer et al. [2017, 2018] study pacing algorithms for budget constraints in both first price auctions and second price auctions.

The techniques of reserve prices and additive boosts have been widely studied in the literature [Amin et al., 2013, Deng et al., 2021a, Golrezaei et al., 2021b, Lavi et al., 2003, Paes Leme et al., 2016, Sandholm and Likhodedov, 2015]. Among these works, Deng et al. [2021b] improve the welfare approximation ratio to $(c+1)/(c+2)$ using boosted auctions with accurate signals about bidders' values in the environment with value maximizers only. Our results are more robust and more general from several aspects: We allow inaccurate signals that approximate bidders' values; Our results hold for a mixture of behavior models in auctions with reserves and/or boosts; We provide guarantees on revenue performance in addition to welfare performance.

Some of our results rely on the assumption that the auto-bidders adopt uniform bidding strategies when the underlying auction is not incentive-compatible for utility-maximizing bidders. In practice, it is usually hard for bidders to adopt and optimize non-uniform bidding strategies, and moreover, uniform bidding has been shown to perform well against optimal non-uniform bidding strategies in ad auctions [Balseiro and Gur, 2019, Bateni et al., 2014, Deng et al., 2020, Feldman and Muthukrishnan, 2008, Feldman et al., 2007].

## 2 Preliminaries

**Position Auctions.** We consider a setting with $n$ bidders bidding simultaneously in $m$ position auctions [Lahaie et al., 2007, Varian, 2007]. In each position auction $j$, we have $s_j$ slots which can be allocated to $s_j$ different bidders. For each bidder $i \in [n]$, we use $v_{i,j}$ to denote the base value of auction $j$ for bidder $i$, and the value of the $k$-th slot in auction $j$ is $v_{i,j} \cdot \mathrm{pos}_{j,k}$. Here $\mathrm{pos}_{j,k} \in \mathbb{R}^+$ is the position normalizer of the $k$-th slot in auction $j$ which does not depend on $i$. Without loss of generality, we assume $\mathrm{pos}_{j,k}$ is decreasing in $k$. For notation convenience, we set $\mathrm{pos}_{j,s_j+1} = 0$ for the non-existing slot. We use $I = (n, m, \{s\}_j, \{v\}_{i,j}, \{\mathrm{pos}\}_{j,k})$ to denote a problem instance and $_{-i}$ to denote the bidders other than bidder $i$. The optimal welfare of the problem instance is defined as

$$\mathrm{Wel(OPT)} = \sum_{j=1}^{m} \sum_{k=1}^{s_j} \mathrm{pos}_{j,k} \cdot k\text{-th highest value of } (v_{1,j}, ..., v_{n,j}).$$

We use $b, p, x$ to denote bidders' bids, payments and allocations. More particularly, $b_{i,j}$ is bidder $i$'s bid in auction $j$. $x_{i,j,k}$ is 1 if bidder $i$ gets the $k$-th slot of auction $j$ and 0 otherwise. $p_{i,j}$ is the price paid by the bidder $i$ in auction $j$. The welfare and revenue of an allocation $x$ and payments $p$ are defined as

$$\mathrm{Wel}(x,p) = \sum_{i=1}^{n} \sum_{j=1}^{m} \sum_{k=1}^{s_j} x_{i,j,k} \cdot v_{i,j} \cdot \mathrm{pos}_{j,k}, \qquad \text{and} \qquad \mathrm{Rev}(x,p) = \sum_{i=1}^{n} \sum_{j=1}^{m} p_{i,j}.$$

For notation convenience, we will use $\mathrm{Wel}_i(x,p) = \sum_{j=1}^{m} \sum_{k=1}^{s_j} x_{i,j,k} \cdot v_{i,j} \cdot \mathrm{pos}_{j,k}$ and $\mathrm{Rev}_i(x,p) = \sum_{j=1}^{m} p_{i,j}$ to represent each bidder's contribution to welfare and revenue.

In this paper, we mainly focus on three auction formats used in position auctions: the Vickrey–Clarke–Groves auction (VCG), the generalized second-price auction (GSP), and the first-price auction (FPA). Their allocation rules are the same: rank bidders by their bids (tie-breaking by bidder indices) and allocate the $k$-th slot to the bidder with the $k$-th highest bid. In the VCG auction each agent pays the externality it imposes on the other the agents, in the GSP auction each agent who is allocated pays the bid of the next highest bidder, and in the FPA each agent who is allocated pays their bid. For notation convenience, we denote by $\hat{b}_{k,j}$ the $k$-th highest bid for $k \in [s_j]$ in auction $j$. Assuming bidder $i$ wins slot $k$ in auction $j$, its payment $p_{i,j}$ in the three auction formats are: (1) VCG: $p_{i,j} = \sum_{\kappa=k+1}^{s_j} \hat{b}_{\kappa,j} \cdot (\mathrm{pos}_{j,\kappa-1} - \mathrm{pos}_{j,\kappa})$, (2) GSP: $p_{i,j} = \hat{b}_{k+1,j} \cdot \mathrm{pos}_{j,k}$, and (3) FPA: $p_{i,j} = b_{i,j} \cdot \mathrm{pos}_{j,k}$. It is not hard to see that, with the same bids, the payments from these three auctions is ranked in an increasing order as VCG, GSP, FPA.

**Reserve Prices and Boosts.** We further consider three auction formats with reserve prices and boosts. We denote by $r_{i,j}$ the reserve price and by $z_{i,j}$ the boost for bidder $i$ in auction $j$. When we have boosts, bidders are ranked by their score $b_{i,j} + z_{i,j}$ and we use $\hat{b}_{k,j}$ to denote the $k$-th highest score for $k \in [s_j]$ in auction $j$. For ease of presentation, we consider lazy reserves such that the slot is not allocated to a bidder if her bid without boosts does not clear her reserve, i.e., $b_{i,j} < r_{i,j}$; our results

continue to hold for eager reserves [Paes Leme et al., 2016]. With reserve prices and boosts, the prices of VCG, GSP and FPA when bidder $i$ gets slot $k$ in auction $j$ become:

- VCG: $p_{i,j} = \sum_{\kappa=k+1}^{s_j} \max(\hat{b}_{\kappa,j} - z_{i,j}, r_{i,j}) \cdot (\text{pos}_{j,\kappa-1} - \text{pos}_{j,\kappa})$;
- GSP: $p_{i,j} = \max(\hat{b}_{k+1,j} - z_{i,j}, r_{i,j}) \cdot \text{pos}_{j,k}$;
- FPA: $p_{i,j} = (\hat{b}_{i,j} - z_{i,j}) \cdot \text{pos}_{j,k}$.

It is worth highlighting that boosts and reserves are fairly different components in auctions. Reserves filter out the candidates whose bids are lower than their reserves and then the remaining candidates are ranked according to their original bids. In contrast, boosts are added to the candidates' ranking scores, and therefore, the candidates are ranked according to their original bids plus their boosts.

**Inaccurate signals.** Our auctions take inaccurate value signals as input and use them as reserve prices and/or boosts. In particular, for $\gamma \in [0, 1]$:

- When the signals are used as boosts, we allow multiplicative errors in both directions and we say boosts are $\gamma$-approx, if for any $i \in [n], j \in [m]$, $z_{i,j} \in [\mu \cdot v_{i,j}, \nu \cdot v_{i,j})$, and $\mu = \gamma \cdot \nu$;
- When the signals are used as reserve prices, we allow multiplicative underestimation errors and say reserve prices are $\gamma$-approx, if for any $i \in [n], j \in [m]$, $r_{i,j} \in [\gamma \cdot v_{i,j}, v_{i,j})$.

Note that we require a one-direction error for reserve signals. One can always convert signals with bounded errors in both directions into signals with only underestimation errors by scaling them down once multiplicative overestimation errors have a finite upper bound.

**Bidders.** We focus on two types of bidders: utility maximizers and value maximizers. A utility maximizer bidder $i$ maximizes $\text{Wel}_i(x, p) - \text{Rev}_i(x, p)$. In contrast, a value maximizer maximizes $\text{Wel}_i(x, p)$ subject to a return on spend constraint $\text{Wel}_i(x, p) \geq \text{Rev}_i(x, p)$. Here, it is without loss of generality to assume the target ratio between return $\text{Wel}_i(x, p)$ and spend $\text{Rev}_i(x, p)$ is 1.

These two types of bidders can be summarized by bidders optimizing the following program

$$\max \quad \text{Wel}_i(x, p) - \lambda_i \cdot \text{Rev}_i(x, p) \tag{1}$$
$$\text{s.t.} \quad \text{Wel}_i(x, p) \geq \text{Rev}_i(x, p).$$

Here $\lambda_i = 0$ corresponds to a value maximizer and $\lambda_i = 1$ corresponds to a utility maximizer. Our results apply for bidders with $\lambda_i \in [0, 1]$. We allow $\lambda_i$ to be different for each bidder and they are unknown to the auctioneer. Here, utility maximizers can be modeled with $\lambda = 1$ because all the auctions we consider are individual rational. As a result, the optimal solution of the objective in (1) is always non-negative, and therefore, the constraint would be irrelevant for utility maximizers.

**Solution Concept.** We consider a solution concept called *undominated bids* which includes the support of Nash equilibrium bids as a subset. Because our results apply to the larger set of undominated bids, they readily hold for refinements such as Nash equilibria, in which our results give price of anarchy bounds [Papadimitriou, 2001]. Following the standard definition of (weak) dominance, for a problem instance $I$ and an auction format, we say a bid vector $b_i = (b_{i,1}, ..., b_{i,m})$ is (weakly) dominated by another bid vector $b'_i = (b'_{i,1}, ..., b'_{i,m})$, if the following two requirements are satisfied:

Let $x, p$ be the allocation and prices induced by $b_i, b'_{-i}$, and $x', p'$ be the allocation and prices induced by $b'_i, b'_{-i}$. Then for any other bidders' bids $b'_{-i}, b'_i$ is at least as good as $b_i$, i.e.,

- both violate constraints: $\text{Wel}_i(x, p) < \text{Rev}_i(x, p)$ and $\text{Wel}_i(x', p') < \text{Rev}_i(x', p')$
- $b_i$ violates constraints while $b'_i$ does not: $\text{Wel}_i(x, p) < \text{Rev}_i(x, p)$ and $\text{Wel}_i(x', p') \geq \text{Rev}_i(x', p')$
- or neither violates constraints, and $b'_i$ yields no worse objective: $\text{Wel}_i(x, p) \geq \text{Rev}_i(x, p)$ and $\text{Wel}_i(x', p') \geq \text{Rev}_i(x', p')$ and $\text{Wel}_i(x', p') - \lambda_i \cdot \text{Rev}_i(x', p') \geq \text{Wel}_i(x, p) - \lambda_i \cdot \text{Rev}_i(x, p)$

There exits $b'_{-i}$ such that $b'_i$ is strictly better than $b_i$, i.e.,

- $b_i$ violates constraints while $b'_i$ does not: $\text{Wel}_i(x, p) < \text{Rev}_i(x, p)$ and $\text{Wel}_i(x', p') \geq \text{Rev}_i(x', p')$

- or neither violates constraints, and $b_i'$ yields strictly better objective: $\text{Wel}_i(x, p) \geq \text{Rev}_i(x, p)$ and $\text{Wel}_i(x', p') \geq \text{Rev}_i(x', p')$ and $\text{Wel}_i(x', p') - \lambda_i \cdot \text{Rev}_i(x', p') > \text{Wel}_i(x, p) - \lambda_i \cdot \text{Rev}_i(x, p)$

We omit the word "weak" for convenience for the rest of the paper. We say $b_i$ is undominated if there is no $b_i'$ dominates $b_i$. Denote the set of all bidders' bids $b = (b_1, ..., b_n)$ by $\Theta$ in which each $b_i$ is undominated for $i \in [n]$ and each bidder is paying at most its welfare.

As uniform bidding is widely adopted in automated bidding strategies, we also consider a solution concept related to it. We say $b_i$ is a uniform bidding for bidder $i$, if $b_{i,j} = v_{i,j} \cdot \delta_i$ for all $j \in [m]$. We use $\Theta_u$ to denote the set of all bidders' bids $b = (b_1, ..., b_n)$ such that each $b_i$ is a uniform bidding and is not dominated by any other uniform bidding and each bidder is paying at most its welfare.

## 3  Main Technical Lemma

We first show a lemma which will be used as the major technical building block for our results. This lemma provides a general framework for proving approximation results on revenue and welfare. Informally, this lemma says that once we can guarantee lower bounds on bidders' bids and apply reserve prices and/or boosts, these immediately lead to lower bound guarantees on revenue and welfare in terms of the optimal welfare. Notice that this lemma continues to hold when we apply reserve prices only (i.e., $\mu = \nu = 0$) or boosts only (i.e., $\beta = 0$). For a better understanding of the lemma statement, conditions number 1, 2 and 5 hold by the definition of the model and conditions number 3 and 4 will be proved hold when using this lemma.

**Lemma 3.1.** *Consider running $m$ position auctions with allocation and pricing rule $\mathcal{A}$. Assuming $\mathcal{A}$ together with bidders' bids $b$ satisfy the following conditions for parameters $\alpha, \beta, \mu, \nu \geq 0$:*

- *For each bidder $i$ and each auction $j$, the reserve satisfies $r_{i,j} \geq \beta \cdot v_{i,j}$ and the boost satisfies $\mu \cdot v_{i,j} \leq z_{i,j} < \nu \cdot v_{i,j}$.*

- *In each position auction, bidders are ranked by their scores $b_{i,j} + z_{i,j}$ and the $k$-th highest score wins the $k$-th slot if $b_{i,j} \geq r_{i,j}$.*

- *If bidder $i$'s value $v_{i,j}$ ranks in top-$s_j$ in auction $j$, bidder $i$ bids at least $\alpha$ times its base value, i.e. $b_{i,j} \geq \alpha \cdot v_{i,j}$.*

- *If bidder $i$ wins slot $k$ in auction $j$, bidder $i$ pays at least the VCG price, i.e. $p_{i,j} \geq \sum_{\kappa=k+1}^{s_j} (pos_{j,\kappa-1} - pos_{j,\kappa}) \cdot \max(\hat{b}_{\kappa,j} - z_{i,j}, r_{i,j})$.*

- *For each bidder, her total payment is at most her total value.*

*We have $Rev(\mathcal{A}(b)) \geq \min\left(\frac{(\alpha+\mu)\beta}{\beta+\nu}, \beta\right) \cdot Wel(OPT)$ and $Wel(\mathcal{A}(b)) \geq \frac{\alpha+\mu}{1+\max(\nu, \alpha+\mu-\beta)} \cdot Wel(OPT)$.*

We give a high-level proof sketch of this lemma. The detailed proof can be found in Appendix A.

**Proof Sketch.** Informally, the first step of the proof is to fractionally partition the auction slots into two parts so that (1) In Part A, $\mathcal{A}(b)$ and OPT agree in allocation; (2) In part B, $\mathcal{A}(b)$ and OPT disagree in allocation. We then lower bound the welfare and the revenue of $\mathcal{A}(b)$ in these two parts separately with different arguments.

In part A, $\mathcal{A}(b)$ and OPT have the same allocation. The welfare of $\mathcal{A}(b)$ is by definition the same as the welfare of OPT. Moreover, the revenue of $\mathcal{A}(b)$ can be lower bounded by $\beta$ times the welfare of OPT, via conditions on reserve prices (the first bullet point of the lemma statement).

In part B, $\mathcal{A}(b)$ and OPT have different allocations as allocated bidders of OPT are not allocated in $\mathcal{A}(b)$. Since we have lower bounds on bids (the third bullet point of the lemma statement), these bidders' bids would give a lower bound on the revenue of $\mathcal{A}(b)$ via VCG pricing. The additional boost $z$ (in the first bullet point of the lemma statement) would escalate this effect and give a lower bound on a linear combination of the welfare and the revenue of $\mathcal{A}(b)$.

By putting these lower bounds together with the condition that $\text{Wel}(\mathcal{A}(b)) \geq \text{Rev}(\mathcal{A}(b))$ (in the fifth bullet point of the lemma statement), we can obtain lower bounds on $\text{Wel}(\mathcal{A}(b))$ and $\text{Rev}(\mathcal{A}(b))$.

# 4 Applications in Auctions

## 4.1 VCG Auctions

In this section, we consider VCG with reserves and boosts. We first show the following lemma about the set of undominated bids $\Theta$. Its proof can be found in Appendix B.

**Lemma 4.1.** *For any problem instance $I$, let $\Theta$ be the set of undominated bids for VCG with reserve $r$ and boost $z$. Assume reserve $r$ satisfies that $r_{i,j} < v_{i,j}$ $\forall i \in [n], j \in [m]$, and boost $z$ satisfies that $z_{i,j} \in [\mu \cdot v_{i,j}, \nu \cdot v_{i,j})$ for some $\nu - \mu \leq 1$ for all $i \in [n], j \in [m]$. For any $b \in \Theta$, we have $b_{i,j} \geq v_{i,j}$, if bidder $i$'s value $v_{i,j}$ ranks in top-$s_j$ in auction $j$.*

With Lemma 4.1, we are ready to state our results on VCG with reserves and boosts. The following three corollaries are the results we get when we apply (1) only reserve prices (2) only boosts (3) reserve prices and boosts together.

Combining Lemma 4.1 with Lemma 3.1 using $\alpha = 1$, $\beta = \gamma$, and $\mu = \nu = 0$:

**Corollary 4.2.** *On any problem instance $I$, VCG with $\gamma$-approx reserve, denoted by $VCG_r^\gamma$, satisfies*

$$Wel(VCG_r^\gamma(b)) \geq \frac{1}{2 - \gamma} \cdot Wel(OPT) \quad and \quad Rev(VCG_r^\gamma(b)) \geq \gamma \cdot Wel(OPT),$$

*for bids $b$ from the undominated bids set $\Theta$.*

Combining Lemma 4.1 with Lemma 3.1 using $\alpha = 1$, $\beta = 0$, $\mu = \gamma/(1 - \gamma)$, and $\nu = 1/(1 - \gamma)$:

**Corollary 4.3.** *On any problem instance $I$, VCG with $\gamma$-approx boosts, denoted by $VCG_b^\gamma$, satisfies*

$$Wel(VCG_b^\gamma(b)) \geq \frac{1}{2 - \gamma} \cdot Wel(OPT)$$

*for bids $b$ from the undominated bids set $\Theta$.*

Combining Lemma 4.1 with Lemma 3.1 using $\alpha = 1$, $\beta = \gamma$, $\mu = \gamma$, and $\nu = 1$:

**Corollary 4.4.** *On any problem instance $I$, VCG with $\gamma$-approx reserve and $\gamma$-approx boost, denoted by $VCG_{r,b}^\gamma$, satisfies*

$$Wel(VCG_{r,b}^\gamma(b)) \geq \frac{\gamma + 1}{2} \cdot Wel(OPT) \quad and \quad Rev(VCG_{r,b}^\gamma(b)) \geq \gamma \cdot Wel(OPT),$$

*for bids $b$ from the undominated bids set $\Theta$.*

## 4.2 Generalized Second-Price Auctions

For GSP, we are able to show a lemma similar to Lemma 4.1 assuming bidders are all value maximizing ($\lambda_i = 0$ $\forall i \in [n]$) and they are uniform bidding. Its proof can be found in Appendix C.

**Lemma 4.5.** *For any problem instance $I$ with $\lambda_i = 0$ $\forall i \in [n]$, let $\Theta_u$ be the set of undominated uniform bids for GSP with with reserve $r$ and boost $z$. Assume reserve $r$ satisfies that $r_{i,j} < v_{i,j}$ $\forall i \in [n], j \in [m]$, and boost $z$ satisfies that for some $\nu - \mu \leq 1$, $z_{i,j} \in [\mu \cdot v_{i,j}, \nu \cdot v_{i,j})$ $\forall i \in [n], j \in [m]$. For any $b \in \Theta_u$, we have $b_{i,j} \geq v_{i,j}$, if bidder $i$'s value $v_{i,j}$ ranks in top-$s_j$ in auction $j$.*

Combining Lemma 4.5 with Lemma 3.1, we can obtain results similar to Corollary 4.2, 4.3, 4.4. We only state the strongest one with both boosts and reserve prices:

**Corollary 4.6.** *On any problem instance $I$ with $\lambda_i = 0$ $\forall i \in [n]$, GSP with $\gamma$-approx reserve and $\gamma$-approx boost, denoted by $GSP_{r,b}^\gamma$, satisfies*

$$Wel(GSP_{r,b}^\gamma) \geq \frac{\gamma + 1}{2} \cdot Wel(OPT) \quad and \quad Rev(GSP_{r,b}^\gamma(b)) \geq \gamma \cdot Wel(OPT),$$

*for uniform bids $b$ from the undominated uniform set $\Theta_u$.*

When there are no restrictions on bidders' bidding behavior, we can obtain the following weaker lemma for GSP with reserves. Its proof can be found in Appendix C.

**Lemma 4.7.** *For any problem instance I, let $\Theta$ be the set of undominated bids for GSP with $\gamma$-approx reserve $r$ and no boosts. For any $b \in \Theta$, we have $b_{i,j} \geq r_{i,j} \geq \gamma \cdot v_{i,j}$, $\forall i \in [n], j \in [m]$.*

Combining Lemma 4.7 with Lemma 3.1 using $\alpha = \gamma$, $\beta = \gamma$, and $\mu = \nu = 0$:

**Corollary 4.8.** *On any problem instance I, GSP with $\gamma$-approx reserve, denoted by $GSP_r^\gamma$, satisfies*

$$Wel(GSP_r^\gamma(b)) \geq \gamma \cdot Wel(OPT) \quad and \quad Rev(GSP_r^\gamma(b)) \geq \gamma \cdot Wel(OPT),$$

*for bids $b$ from the undominated bids set $\Theta$.*

### 4.3 First-Price Auctions

For FPA, if we restrict that bidders are only value maximizers and they are uniform bidding, we know from prior work that optimal welfare and revenue can be achieved without reserves and boosts. Its proof can be found in Appendix D.

**Theorem (Deng et al. [2021b]).** *On any problem instance I with $\lambda_i = 0$ $\forall i \in [n]$, FPA has*

$$Wel(FPA(b)) = Rev(FPA(b)) = Wel(OPT),$$

*for uniform bids $b$ from the undominated uniform set $\Theta_u$.*

When there are no restrictions on bidders, we get similar results as GSP with the following lemma.

**Lemma 4.9.** *For any problem instance I, let $\Theta$ be the set of undominated bids for FPA with $\gamma$-approx reserve $r$ and no boosts. For any $b \in \Theta$, we have $b_{i,j} \geq r_{i,j} \geq \gamma \cdot v_{i,j}$, $\forall i \in [n], j \in [m]$.*

Combining Lemma 4.9 with Lemma 3.1 using $\alpha = \gamma$, $\beta = \gamma$, and $\mu = \nu = 0$:

**Corollary 4.10.** *On any problem instance I, FPA with $\gamma$-approx reserve ($FPA_r^\gamma$) has*

$$Wel(FPA_r^\gamma(b)) \geq \gamma \cdot Wel(OPT) \quad and \quad Rev(FPA_r^\gamma(b)) \geq \gamma \cdot Wel(OPT),$$

*for bids $b$ from the undominated bids set $\Theta$.*

## 5 Experiments

In this section, we derive semi-synthetic data from real auction data of a major search engine to validate our theoretical findings concerning VCG auctions. VCG auctions provide a clean environment for us to validate our findings as uniform bidding is a best response for each value maximizer [Aggarwal et al., 2019] while computing best response in GSP auctions is a highly non-trivial task.

Observe that when the bidders are symmetric such that their valuation distributions are identically and independently distributed, both optimal efficiency and optimal revenue are achieved in any symmetric equilibrium, and thus, no efficiency or revenue improvement can be observed by applying our mechanisms. Therefore, we use real ad auction data for capturing variation across bidders. For the experimental purpose, we simulate VCG auctions with bids from value maximizers only, as utility maximizers do not respond to either boosts or reserve prices in VCG mechanisms. Instead of using real return on ad spend targets for value maximizers, we generate artificial targets to exclude any practical noises from the real system. We emphasize that the main objective for our empirical study is to validate our theoretical findings rather than investigating the efficiency and/or revenue potentials on real systems actually implemented in practice. Many practical aspects often need to be taken care of in the design of real systems, which would never be considered in theory.

**Simulation Procedures.** To properly evaluate the efficiency and revenue of a new mechanism, after the generation of the dataset, we first pre-train the (uniform) bid multipliers $\delta$ for value maximizers in 25 iterations without reserve prices and boosts to obtain an equilibrium as a starting point. We then simulate the response of value maximizers by gradient descent on their bid multipliers in log space until convergence [Aggarwal et al., 2019, Nesterov, 2013]. Formally, let $\delta_{i,t}$ be the bid multiplier for value maximizer $i$ in iteration $t$. In addition, let $Wel_{i,t}$ be value maximizer $i$'s total received value in iteration $t$, and let $Rev_{i,t}$ be her total payment in iteration $t$. Then, in iteration $t + 1$, the value maximizer $i$'s bid multiplier is updated by $\log \delta_{i,t+1} = (1 - \eta_t) \cdot \log \delta_{i,t} + \eta_t \cdot \log \frac{Wel_{i,t}}{Rev_{i,t}}$, where $\eta_t \in (0,1)$ is a properly chosen learning rate for $t$-th iteration. Intuitively, when $Rev_{i,t} < Wel_{i,t}$,

value maximizer $i$'s bid multiplier increases for the next iteration; otherwise, her bid multiplier decreases for the next iteration. After obtaining a starting point, we simulate another 25 iterations for auctions with reserve prices and/or boosts. In this way, we can observe both the initial impact of adding reserve prices and/or boosts and how the impact changes over time during value maximizers' response until convergence.

**Reserve Prices and Boosts.** In addition to the baseline in which we continue to use auctions without reserve prices, we experiment with boosts and reserve prices using signals with different approximation factors $\gamma$. For each bidder $i$ in each auction $j$, we will set reserve prices or give boosts based on $\gamma$ in different treatments. Here, let $s_{i,j}^{\gamma}$ be a random variable independently sampled from a Gaussian distribution with mean $(1+\gamma)/2$ and standard error 0.01, truncated within $[\gamma, 1]$. In treatment reserve-$\gamma$, we set the reserve price as $s_{i,j}^{\gamma} \cdot v_{i,j}$. In treatment boost-$\gamma$, we add an additive boost $\frac{1}{1-\gamma} \cdot s_{i,j}^{\gamma} \cdot v_{i,j}$ as suggested. Finally, in treatment boost-reserve-$\gamma$, we set the reserve price as $s_{i,j}^{\gamma} \cdot v_{i,j}$ and additionally give an additive boost $\frac{1}{1-\gamma} \cdot s_{i,j}^{\gamma} \cdot v_{i,j}$. All metrics we report are relative to the gap between the baseline (i.e., without treatment) and the optimal solution. More precisely, let $\kappa_{\text{init}}$ be the initial welfare (revenue) without reserve prices or boosts, $\kappa_{\text{e}}$ be the welfare (revenue) under treatment $e$, and $\kappa(\text{OPT})$ be the optimal welfare (revenue). Then the reported percentage is computed by $(\kappa_{\text{e}} - \kappa_{\text{init}})/(\kappa(\text{OPT}) - \kappa_{\text{init}})$.

## 5.1 Experimental Results

Figure 1a reports the trend of welfare performance in VCG auctions under different treatments with one run of the experiment. Observe that both variants of reserve prices have neutral initial impact on welfare before value maximizers start to respond. After the value maximizers start to respond, they adjust their bid multipliers towards a better allocation, resulting in a positive impact on welfare. As the reserve prices become more precise (i.e., when $\gamma$ is closer to 1), the welfare impact is larger, which confirms our theoretical results of Corollary 4.2. In contrast, both variants of boosts have positive initial impact on welfare. However, as the value maximizers start to respond, the welfare impact starts to decrease but the final impact after convergence is still positive, confirming our theoretical results of Corollary 4.3. Interestingly, when treatment reserve-$\gamma$ and treatment boost-$\gamma$ share the same $\gamma$, their final impact on revenue after response are close to each other, as predicted by the same welfare bounds from Corollary 4.2 and 4.3. Moreover, a combination of boosts and reserve prices outperforms the treatments with either reserve prices only or boosts only, which validates Corollary 4.4 that enjoys a better bound than Corollary 4.2 and 4.3.

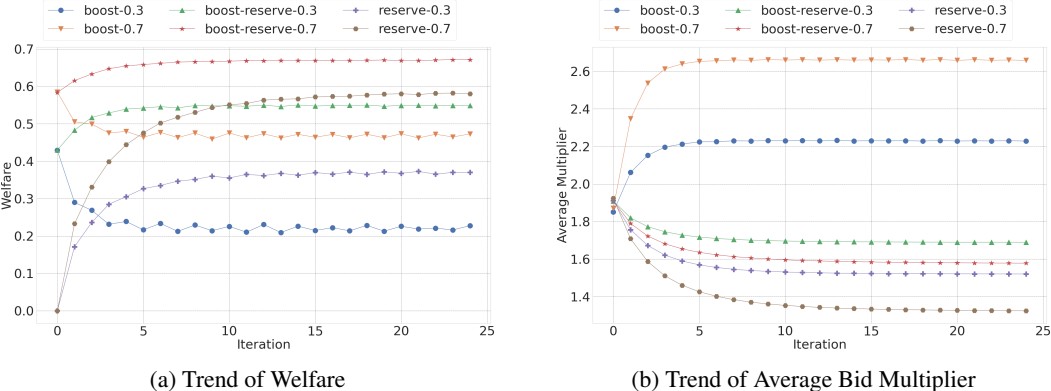

(a) Trend of Welfare                 (b) Trend of Average Bid Multiplier

Figure 1: Convergence during response per iteration.

Figure 1b demonstrates the trend of average (uniform) bid multipliers for value maximizers under different treatments in VCG auctions with one run of the experiment. We observe that the average multiplier decreases during response for both treatments with reserve prices only, leading to a better welfare performance post response as shown in Figure 1a. Moreover, as the reserve prices become more precise, the average multiplier is lower after convergence. Intuitively, the welfare is maximized when all the value maximizers adopt a bid multiplier 1 so that the auction results in a ranking according to their values. Therefore, the bid multipliers getting close to 1 after response is the main driving factor behind the increase of welfare for treatments with reserve prices only. On the other

hand, the average multiplier increases during response for both treatments with boosts only, leading to a worse welfare performance post response as shown in Figure 1a. A combination of reserve prices and boosts induces a mild decrease of the average multiplier, resulting in a rather stable welfare performance during value maximizers' response.

|  | Signal | Reserve Only | Boost Only | Boost + Reserve |
|---|---|---|---|---|
| Welfare Lift | $\gamma = 0.3$ | $37.4\% \pm 3.2\%$ | $23.0\% \pm 3.4\%$ | $55.4\% \pm 3.3\%$ |
|  | $\gamma = 0.5$ | $47.9\% \pm 2.9\%$ | $33.6\% \pm 3.2\%$ | $63.2\% \pm 3.1\%$ |
|  | $\gamma = 0.7$ | $58.6\% \pm 3.3\%$ | $47.7\% \pm 3.2\%$ | $67.6\% \pm 3.2\%$ |
| Revenue Lift | $\gamma = 0.3$ | $28.2\% \pm 1.7\%$ | $23.7\% \pm 2.2\%$ | $44.9\% \pm 1.9\%$ |
|  | $\gamma = 0.5$ | $39.4\% \pm 1.9\%$ | $35.3\% \pm 2.5\%$ | $52.9\% \pm 1.8\%$ |
|  | $\gamma = 0.7$ | $54.1\% \pm 1.8\%$ | $51.2\% \pm 3.0\%$ | $59.9\% \pm 2.2\%$ |

Table 2: Welfare and revenue lifts for different treatments after convergence.

We conduct 10 runs of repeated experiments and Table 2 shows the welfare and revenue impact after convergence with 95% confidence intervals. First, notice that treatments with reserve prices only have better welfare and revenue lifts than treatments with boosts when they share the same $\gamma$. Moreover, a combination of boosts and reserve prices leads to further improvements in both revenue and welfare. For all variants, as the signals becomes more precise (i.e., $\gamma$ is closer to 1), the impacts are larger. Note that in an environment with value maximizers only, if all value maximizers could always hit their target spends after convergence, the improved ratios for welfare and revenue should be the same. However, due to the discontinuity of the bidding landscape, there are value maximizers who cannot hit their targets in practice. As a result, we witness difference between welfare improvement and revenue improvement even after convergence.

## 6 Conclusion

In this paper, we provide both theoretical and empirical evidence to demonstrate that introducing properly chosen reserve prices and/or boosts can improve both revenue and welfare in the auto-bidding world. Our results are robust to bidders' behavior models as well as inaccurate signals that approximate bidders' values. One limitation of our results is regarding the requirement of reserve prices not exceeding bidder values. Although one can avoid this by scaling down the reserve prices, it does require additional knowledge from the auction designer on an upper limit of the inaccurate signals. It is an intriguing question to address such an issue in a more robust way.

## 7 Acknowledgement

Thanks are due to Mohammad Mahdian, Aranyak Mehta, Renato Paes Leme, and Ying Wang for helpful discussions, and for their comments and suggestions.

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
