# A Missing Proofs of Section 3

*Proof.* (of Lemma 3.1)

For notation convenience, we define $S_{j,k}$ and $S_{j,k}^{\text{OPT}}$ to be the set of bidders who get allocated to any of the top-$k$ slots of auction $j$ in $\mathcal{A}$ and OPT, respectively.

The optimal welfare can be written, by exchanging summations, as

$$\text{Wel(OPT)} = \sum_{j=1}^{m} \sum_{k=1}^{s_j} \sum_{i \in S_{j,k}^{\text{OPT}}} (\text{pos}_{j,k} - \text{pos}_{j,k+1}) \cdot v_{i,j},$$

and the welfare of $\mathcal{A}$ on bids $b$ can be written as

$$\text{Wel}(\mathcal{A}(b)) = \sum_{j=1}^{m} \sum_{k=1}^{s_j} \sum_{i \in S_{j,k}} (\text{pos}_{j,k} - \text{pos}_{j,k+1}) \cdot v_{i,j}.$$

We start by lower bounding $\text{Rev}(\mathcal{A}(b))$. For each auction $j$, by the condition on payments and rearranging terms, we get

$$\text{Rev}(\mathcal{A}(b)) \geq \sum_{j=1}^{m} \sum_{k=1}^{s_j} (\text{pos}_{j,k} - \text{pos}_{j,k+1}) \cdot \sum_{i \in S_{j,k}} \max(\hat{b}_{k+1,j} - z_{i,j}, v_{i,j} \cdot \beta).$$

For $k$-th slot in auction $j$, we partition the revenue into two components:

$$\sum_{i \in S_{j,k}} \max(\hat{b}_{k+1,j} - z_{i,j}, v_{i,j} \cdot \beta)$$

$$= \sum_{i \in S_{j,k} \cap S_{j,k}^{\text{OPT}}} \max(\hat{b}_{k+1,j} - z_{i,j}, v_{i,j} \cdot \beta) + \sum_{i \in S_{j,k} \setminus S_{j,k}^{\text{OPT}}} \max(\hat{b}_{k+1,j} - z_{i,j}, v_{i,j} \cdot \beta)$$

For the first component, we simply have

$$\sum_{i \in S_{j,k} \cap S_{j,k}^{\text{OPT}}} \max(\hat{b}_{k+1,j} - z_{i,j}, v_{i,j} \cdot \beta) \geq \sum_{i \in S_{j,k} \cap S_{j,k}^{\text{OPT}}} v_{i,j} \cdot \beta. \tag{2}$$

For the second component, consider any $i \in S_{j,k}^{\text{OPT}} \setminus S_{j,k}$. We know in $\mathcal{A}$, bidder $i$ ranks out of top-$k$ in auction $j$. This means $\hat{b}_{k+1,j}$ is at least as high as bidder $i$'s score which is at least $v_{i,j} \cdot \alpha + z_{i,j}$. Since $|S_{j,k}^{\text{OPT}}| = |S_{j,k}| = k$, we know $|S_{j,k} \setminus S_{j,k}^{\text{OPT}}| = k - |S_{j,k} \cap S_{j,k}^{\text{OPT}}| = |S_{j,k}^{\text{OPT}} \setminus S_{j,k}|$. Therefore,

$$\sum_{i \in S_{j,k} \setminus S_{j,k}^{\text{OPT}}} \max(\hat{b}_{k+1,j} - z_{i,j}, v_{i,j} \cdot \beta)$$

$$\geq |S_{j,k} \setminus S_{j,k}^{\text{OPT}}| \cdot \hat{b}_{k+1,j} - \sum_{i \in S_{j,k} \setminus S_{j,k}^{\text{OPT}}} z_{i,j}$$

$$= |S_{j,k}^{\text{OPT}} \setminus S_{j,k}| \cdot \hat{b}_{k+1,j} - \sum_{i \in S_{j,k} \setminus S_{j,k}^{\text{OPT}}} z_{i,j}$$

$$\geq \left( \sum_{i \in S_{j,k}^{\text{OPT}} \setminus S_{j,k}} v_{i,j} \cdot \alpha + z_{i,j} \right) - \left( \sum_{i \in S_{j,k} \setminus S_{j,k}^{\text{OPT}}} z_{i,j} \right)$$

$$\geq \left( \sum_{i \in S_{j,k}^{\text{OPT}} \setminus S_{j,k}} v_{i,j} \cdot (\alpha + \mu) \right) - \left( \sum_{i \in S_{j,k} \setminus S_{j,k}^{\text{OPT}}} v_{i,j} \cdot \nu \right) \tag{3}$$

We also have

$$\sum_{i \in S_{j,k} \setminus S_{j,k}^{\text{OPT}}} \max(\hat{b}_{k+1,j} - z_{i,j}, v_{i,j} \cdot \beta) \geq \sum_{i \in S_{j,k} \setminus S_{j,k}^{\text{OPT}}} v_{i,j} \cdot \beta. \tag{4}$$

Therefore, by inequalities (3) and (4), we get

$$\sum_{i \in S_{j,k} \setminus S_{j,k}^{\text{OPT}}} \max(\hat{b}_{k+1,j} - z_{i,j}, v_{i,j} \cdot \beta)$$

$$= \frac{\beta}{\beta + \nu} \left( \sum_{i \in S_{j,k} \setminus S_{j,k}^{\text{OPT}}} \max(\hat{b}_{k+1,j} - z_{i,j}, v_{i,j} \cdot \beta) \right) + \frac{\nu}{\beta + \nu} \left( \sum_{i \in S_{j,k} \setminus S_{j,k}^{\text{OPT}}} \max(\hat{b}_{k+1,j} - z_{i,j}, v_{i,j} \cdot \beta) \right)$$

$$\geq \left( \sum_{i \in S_{j,k}^{\text{OPT}} \setminus S_{j,k}} v_{i,j} \cdot (\alpha + \mu) \cdot \frac{\beta}{\beta + \nu} \right) - \left( \sum_{i \in S_{j,k} \setminus S_{j,k}^{\text{OPT}}} v_{i,j} \cdot \nu \cdot \frac{\beta}{\beta + \nu} \right) + \left( \sum_{i \in S_{j,k} \setminus S_{j,k}^{\text{OPT}}} v_{i,j} \cdot \beta \cdot \frac{\nu}{\nu + \beta} \right)$$

$$= \sum_{i \in S_{j,k}^{\text{OPT}} \setminus S_{j,k}} v_{i,j} \cdot \frac{(\alpha + \mu)\beta}{\beta + \nu}.$$

Together with inequality (2), we have

$$\text{Rev}(\mathcal{A}(b)) \geq \sum_{j=1}^{m} \sum_{k=1}^{s_j} (\text{pos}_{j,k} - \text{pos}_{j,k+1}) \cdot \sum_{i \in S_{j,k}} \max(\hat{b}_{k+1,j} - z_{i,j}, v_{i,j} \cdot \beta)$$

$$\geq \sum_{j=1}^{m} \sum_{k=1}^{s_j} (\text{pos}_{j,k} - \text{pos}_{j,k+1}) \left( \sum_{i \in S_{j,k} \cap S_{j,k}^{\text{OPT}}} v_{i,j} \cdot \beta + \sum_{i \in S_{j,k}^{\text{OPT}} \setminus S_{j,k}} v_{i,j} \cdot \frac{(\alpha + \mu)\beta}{\beta + \nu} \right)$$

$$\geq \sum_{j=1}^{m} \sum_{k=1}^{s_j} (\text{pos}_{j,k} - \text{pos}_{j,k+1}) \sum_{i \in S_{j,k}^{\text{OPT}}} v_{i,j} \cdot \min \left( \frac{(\alpha + \mu)\beta}{\beta + \nu}, \beta \right)$$

$$= \min \left( \frac{(\alpha + \mu)\beta}{\beta + \nu}, \beta \right) \cdot \text{Wel}(\text{OPT}).$$

Now we lower bound $\text{Wel}(\mathcal{A}(b))$. By the third condition, we know $\text{Wel}(\mathcal{A}(b)) \geq \text{Rev}(\mathcal{A}(b))$. Here we prove a slightly different lower bound on $\text{Rev}(\mathcal{A}(b))$ using inequalities (2) and (3):

$$\text{Rev}(\mathcal{A}(b))$$

$$\geq \sum_{j=1}^{m} \sum_{k=1}^{s_j} (\text{pos}_{j,k} - \text{pos}_{j,k+1}) \cdot \sum_{i \in S_{j,k}} \max(\hat{b}_{k+1,j} - z_{i,j}, v_{i,j} \cdot \beta)$$

$$\geq \sum_{j=1}^{m} \sum_{k=1}^{s_j} (\text{pos}_{j,k} - \text{pos}_{j,k+1}) \left( \sum_{i \in S_{j,k} \cap S_{j,k}^{\text{OPT}}} v_{i,j} \cdot \beta + \sum_{i \in S_{j,k}^{\text{OPT}} \setminus S_{j,k}} v_{i,j} \cdot (\alpha + \mu) - \sum_{i \in S_{j,k} \setminus S_{j,k}^{\text{OPT}}} v_{i,j} \cdot \nu \right)$$

$$\tag{5}$$

There are two cases. If $\alpha + \mu - \beta \leq \nu$, we have

$$\text{Wel}(\mathcal{A}(b))$$

$$\geq \frac{\nu}{1+\nu} \cdot \text{Wel}(\mathcal{A}(b)) + \frac{1}{1+\nu} \cdot \text{Rev}(\mathcal{A}(b))$$

$$\geq \sum_{j=1}^{m} \sum_{k=1}^{s_j} (\text{pos}_{j,k} - \text{pos}_{j,k+1}) \left( \sum_{i \in S_{j,k} \cap S_{j,k}^{\text{OPT}}} v_{i,j} + \sum_{i \in S_{j,k} \setminus S_{j,k}^{\text{OPT}}} v_{i,j} \right) \cdot \frac{\nu}{1+\nu}$$

$$+ \sum_{j=1}^{m} \sum_{k=1}^{s_j} (\text{pos}_{j,k} - \text{pos}_{j,k+1})$$

$$\cdot \left( \sum_{i \in S_{j,k} \cap S_{j,k}^{\text{OPT}}} v_{i,j} \cdot \beta + \sum_{i \in S_{j,k}^{\text{OPT}} \setminus S_{j,k}} v_{i,j} \cdot (\alpha + \mu) - \sum_{i \in S_{j,k} \setminus S_{j,k}^{\text{OPT}}} v_{i,j} \cdot \nu \right) \cdot \frac{1}{1+\nu}$$

(by the definition of $\text{Wel}(\mathcal{A}(b))$ and inequality (5))

$$= \sum_{j=1}^{m} \sum_{k=1}^{s_j} (\text{pos}_{j,k} - \text{pos}_{j,k+1}) \left( \sum_{i \in S_{j,k} \cap S_{j,k}^{\text{OPT}}} v_{i,j} \cdot \frac{\beta+\nu}{1+\nu} + \sum_{i \in S_{j,k}^{\text{OPT}} \setminus S_{j,k}} v_{i,j} \cdot \frac{\alpha+\mu}{1+\nu} \right)$$

$$\geq \frac{\alpha+\mu}{1+\nu} \cdot \sum_{j=1}^{m} \sum_{k=1}^{s_j} (\text{pos}_{j,k} - \text{pos}_{j,k+1}) \left( \sum_{i \in S_{j,k} \cap S_{j,k}^{\text{OPT}}} v_{i,j} + \sum_{i \in S_{j,k}^{\text{OPT}} \setminus S_{j,k}} v_{i,j} \right)$$

$$= \frac{\alpha+\mu}{1+\max(\nu, \alpha+\mu-\beta)} \cdot \text{Wel}(\text{OPT}).$$

If $\alpha + \mu - \beta > \nu$, we have

$$\text{Wel}(\mathcal{A}(b))$$

$$\geq \frac{\alpha+\mu-\beta}{1+\alpha+\mu-\beta} \cdot \text{Wel}(\mathcal{A}(b)) + \frac{1}{1+\alpha+\mu-\beta} \cdot \text{Rev}(\mathcal{A})$$

$$\geq \frac{\alpha+\mu}{1+\alpha+\mu-\beta} \cdot \sum_{j=1}^{m} \sum_{k=1}^{s_j} (\text{pos}_{j,k} - \text{pos}_{j,k+1})$$

$$\cdot \left( \sum_{i \in S_{j,k} \cap S_{j,k}^{\text{OPT}}} v_{i,j} + \sum_{i \in S_{j,k}^{\text{OPT}} \setminus S_{j,k}} v_{i,j} + \sum_{i \in S_{j,k} \setminus S_{j,k}^{\text{OPT}}} v_{i,j} \cdot \frac{\alpha+\mu-\beta-\nu}{\alpha+\mu} \right)$$

(by the definition of $\text{Wel}(\mathcal{A}(b))$ and inequality (5))

$$\geq \frac{\alpha+\mu}{1+\alpha+\mu-\beta} \cdot \sum_{j=1}^{m} \sum_{k=1}^{s_j} (\text{pos}_{j,k} - \text{pos}_{j,k+1}) \left( \sum_{i \in S_{j,k} \cap S_{j,k}^{\text{OPT}}} v_{i,j} + \sum_{i \in S_{j,k}^{\text{OPT}} \setminus S_{j,k}} v_{i,j} \right)$$

$$= \frac{\alpha+\mu}{1+\max(\nu, \alpha+\mu-\beta)} \cdot \text{Wel}(\text{OPT}).$$

$\square$

## B  Missing Proofs of Section 4.1

*Proof.* (of Lemma 4.1) We prove by contradiction. Suppose that there exists some $b \in \Theta$ such that $b_{i',j'} < v_{i',j'}$ for some $i' \in [n], j' \in [m]$, and $v_{i',j'}$ ranks in top-$s_{j'}$ in auction $j'$. Define $b'_{i'}$ to be the same as $b_{i'}$ except $b'_{i',j'} = v_{i',j'}$. We want to show $b'_{i'}$ dominates $b_{i'}$.

We start with proving the first requirement for $b'_{i'}$ dominating $b_{i'}$. For any $b'_{-i'}$, let $x, p$ be the allocation of bids $(b_{i'}, b'_{-i'})$, and $x', p'$ be the allocation of bids $(b'_{i'}, b'_{-i'})$. Since bids are the same in

auctions other than $j'$, we know that $x_{i',j,k} = x'_{i',j,k}$ and $p_{i',j} = p'_{i',j}$ for any $j \neq j'$ and $k \in [s_j]$. Therefore, the payment difference for bidder $i$ is

$$\sum_{j=1}^{m}(p'_{i',j} - p_{i',j}) = p'_{i',j'} - p_{i',j'}.$$

and the welfare difference for bidder $i$ is

$$\sum_{j=1}^{m}\sum_{k=1}^{s_j}(x'_{i',j,k} - x_{i',j,k}) \cdot v_{i',j} \cdot \text{pos}_{j,k} = \sum_{k=1}^{s_{j'}}(x'_{i',j',k} - x_{i',j',k}) \cdot v_{i',j'} \cdot \text{pos}_{j',k}$$

We want to show that

$$0 \leq p'_{i',j'} - p_{i',j'} \leq \sum_{k=1}^{s_{j'}}(x'_{i',j',k} - x_{i',j',k}) \cdot v_{i',j'} \cdot \text{pos}_{j',k}.$$

There are two cases. The first case is that bidder $i'$ does not get allocated in auction $j'$ in allocation $x$. In this case, we only need to show $0 \leq p'_{i',j'} \leq \sum_{k=1}^{s_{j'}} x'_{i',j',k} \cdot v_{i',j'} \cdot \text{pos}_{j',k}$. Notice that $b_{i',j'} = v_{i',j'}$. And we can derive from the payment definition of VCG that the payment is non-negative and at most bid multiplied by the position normalizer. Therefore, we have $0 \leq p'_{i',j'} \leq \sum_{k=1}^{s_{j'}} x'_{i',j',k} \cdot v_{i',j'} \cdot \text{pos}_{j',k}$.

In the second case, bidder $i'$ gets allocated to some slot $q$ in auction $j'$ in allocation $x$. Since $b'_{i',j'} = v_{i',j'} > b_{i',j'}$, we know bidder $i'$ gets allocated to some slot $q'$ in auction $j'$ in allocation $x$ and we have $q' \leq q$. Now we can write the welfare difference

$$\sum_{k=1}^{s_{j'}}(x'_{i',j',k} - x_{i',j',k}) \cdot v_{i',j'} \cdot \text{pos}_{j',k} = v_{i',j'} \cdot (\text{pos}_{j',q'} - \text{pos}_{j',q}).$$

On the other hand, the payment difference $p'_{i',j'} - p_{i',j'}$ can be upper bounded by $(\text{pos}_{j',q'} - \text{pos}_{j',q})$ multiplied by the maximum of $r_{i',j'}$ and $v_{i',j'}$. And we also know that $r_{i',j'} \leq v_{i',j'}$. Therefore,

$$0 \leq p'_{i',j'} - p_{i',j'} \leq (\text{pos}_{j',q'} - \text{pos}_{j',q}) \cdot v_{i',j'}$$

To sum up, we get that the payment difference is at most the welfare difference, and they are non-negative, i.e.

$$0 \leq \sum_{j=1}^{m}(p'_{i',j} - p_{i',j}) \leq \sum_{j=1}^{m}\sum_{k=1}^{s_j}(x'_{i',j,k} - x_{i',j,k}) \cdot v_{i',j} \cdot \text{pos}_{j,k}.$$

Since $\lambda_{i'} \in [0, 1]$, this implies that the objective is higher in $(x', p')$ than in $(x, p)$, i.e.

$$\sum_{j=1}^{m}\sum_{k=1}^{s_j} x'_{i',j,k} \cdot v_{i',j} \cdot \text{pos}_{j,k} - \lambda_{i'} \cdot \sum_{j=1}^{m} p'_{i',j} \geq \sum_{j=1}^{m}\sum_{k=1}^{s_j} x_{i',j,k} \cdot v_{i',j} \cdot \text{pos}_{j,k} - \lambda_{i'} \cdot \sum_{j=1}^{m} p_{i',j}.$$

And the constraint is better satisfied, i.e.

$$\sum_{j=1}^{m}\sum_{k=1}^{s_j} x'_{i',j,k} \cdot v_{i',j} \cdot \text{pos}_{j,k} - \sum_{j=1}^{m} p'_{i',j} \geq \sum_{j=1}^{m}\sum_{k=1}^{s_j} x_{i',j,k} \cdot v_{i',j} \cdot \text{pos}_{j,k} - \sum_{j=1}^{m} p_{i',j}.$$

They together imply the first requirement for $b'_{i'}$ dominating $b_{i'}$ is satisfied.

Now we continue to prove the second requirement for $b'_{i'}$ dominating $b_{i'}$. Consider the following $b'_{-i'}$:

- For auction $j'$, and bidder $i \neq i'$ with value ranks higher than bidder $i$ in auction $j'$, set $b'_{i,j'} = v_{i,j'} + z_{i',j'}$.

- For auction $j'$, and bidder $i \neq i'$ with value ranks lower than bidder $i$ in auction $j'$, set $b'_{i,j'} = \frac{v_{i,j'} + b_{i',j'}}{2} + z_{i',j'} - z_{i,j'}$.

- For other auction $j \neq j'$ and bidder $i \neq i'$, set $b'_{i,j} = 0$.

Again, let $x, p$ be the allocation of bids $(b_{i'}, b'_{-i'})$, and $x', p'$ be the allocation of bids $(b'_{i'}, b'_{-i'})$. For auction $j'$, let bidder $i'$'s value ranks at $q$. With bid $b'_{i',j'} = v_{i',j'}$, it's easy to check that bidder $i'$'s score ranks also at $q$. On the other hand, with bid $b_{i',j'} < v_{i',j'}$, bidder $i'$ ranks the last. So in $x'$ the allocation is better and it's easy to see that the welfare improvement in $x'$ is larger than the payment increase. For other auction $j \neq j'$, bidder $i'$ gets the same allocation and payments in $(x, p)$ and $(x', p')$. Therefore, we know

$$\text{Wel}_{i'}(x', p') - \lambda_{i'} \cdot \text{Rev}_{i'}(x', p') > \text{Wel}_{i'}(x, p) - \lambda_{i'} \cdot \text{Rev}_{i'}(x, p).$$

In auction $j \neq j'$, competing bids are 0 and therefore, we bound payments by reserves:

$$p'_{i',j} \leq \sum_{k=1}^{s_j} x'_{i',j,k} \cdot r_{i',j} \cdot \text{pos}_{j,k} \leq \sum_{k=1}^{s_j} x'_{i',j,k} \cdot v_{i',j}.$$

In auction $j'$, we bound payments by the bid $b'_{i',j'} = v'_{i',j'}$:

$$p'_{i',j'} \leq \sum_{k=1}^{s'_j} x'_{i',j',k} \cdot b_{i',j'} \cdot \text{pos}_{j',k} = \sum_{k=1}^{s'_j} x'_{i',j',k} \cdot v_{i',j'}.$$

They together give $\text{Rev}_{i'}(x', p') \leq \text{Wel}_{i'}(x', p')$.

With $\text{Wel}_{i'}(x', p') - \lambda_{i'} \cdot \text{Rev}_{i'}(x', p') > \text{Wel}_{i'}(x, p) - \lambda_{i'} \cdot \text{Rev}_{i'}(x, p)$ and $\text{Rev}_{i'}(x', p') \leq \text{Wel}_{i'}(x', p')$, we know the second requirement for $b'_{i'}$ dominating $b_{i'}$ is satisfied. Now we have $b'_{i'}$ dominates $b_{i'}$. Thus, we get a contradiction. □

# C   Missing Proofs of Section 4.2

*Proof.* (of Lemma 4.5) We prove by contradiction. Suppose that there exists some uniform bids $b \in \Theta_u$ such that $b_{i',j'} < v_{i',j'}$ for some $i' \in [n], j' \in [m]$. Since $b_{i'}$ is a uniform bidding, we can write $b_{i',j} = \delta_{i'} \cdot v_{i',j}, \forall j \in [m]$, and we know $\delta_{i'} < 1$.

Define $b'_{i'}$ to be a uniform bidding such that $b'_{i',j} = v_{i',j}, \forall j \in [m]$. We want to show $b'_{i'}$ dominates $b_{i'}$.

We start with proving the first requirement for $b'_{i'}$ dominating $b_{i'}$. For any $b'_{-i'}$, let $x, p$ be the allocation of bids $(b_{i'}, b'_{-i'})$, and $x', p'$ be the allocation of bids $(b'_{i'}, b'_{-i'})$. Since $b_{i',j} \geq b'_{i',j}, \forall j \in [m]$, we know that bidder $i'$ is ranked no worse in $x$ than in $x'$ for each auction $j \in [m]$. This implies that

$$\text{Wel}_{i'}(x', p') \geq \text{Wel}_{i'}(x, p)$$

On the other hand, we know that in GSP with reserve, the payment in each auction for each bidder is at most its bid multiplied by the position normalizer. Therefore, we have

$$\text{Rev}_{i'}(x', p') \leq \sum_{j=1}^{m} \sum_{k=1}^{s_j} x'_{i',j,k} \cdot b'_{i',j} \cdot \text{pos}_{j,k} \leq \sum_{j=1}^{m} \sum_{k=1}^{s_j} x'_{i',j,k} \cdot v_{i',j} \cdot \text{pos}_{j,k} = \text{Wel}_{i'}(x', p').$$

Now we continue to prove the second requirement for $b'_{i'}$ dominating $b_{i'}$. Consider the following $b'_{-i'}$:

- For auction $j'$, and bidder $i \neq i'$ with value ranks higher than bidder $i$ in auction $j'$, set $b'_{i,j'} = v_{i,j'} + z_{i',j'}$.

- For auction $j'$, and bidder $i \neq i'$ with value ranks lower than bidder $i$ in auction $j'$, set $b'_{i,j'} = \frac{v_{i,j'} + b_{i',j'}}{2} + z_{i',j'} - z_{i,j'}$.

- For other auction $j \neq j'$ and bidder $i \neq i'$, set $b'_{i,j}$ to be consistent with $b'_{i,j'}$ according to uniform bidding.

Again, let $x, p$ be the allocation of bids $(b_{i'}, b'_{-i'})$, and $x', p'$ be the allocation of bids $(b'_{i'}, b'_{-i'})$. For auction $j'$, let bidder $i'$'s value ranks at $q$. With bid $b'_{i',j'} = v_{i',j'}$, it's easy to check that bidder $i'$'s score ranks also at $q$. On the other hand, with bid $b_{i',j'} < v_{i',j'}$, bidder $i'$ ranks the last. So in $x'$ the allocation is better and it's easy to see that the welfare improvement in $x'$ is larger than the payment increase. For other auction $j \neq j'$, bidder $i'$ gets the same allocation and payments in $(x, p)$ and $(x', p')$. Therefore, we know

$$\text{Wel}_{i'}(x', p') - \lambda_{i'} \cdot \text{Rev}_{i'}(x', p') > \text{Wel}_{i'}(x, p) - \lambda_{i'} \cdot \text{Rev}_{i'}(x, p).$$

And we have already shown $\text{Rev}_{i'}(x', p') \leq \text{Wel}_{i'}(x', p')$ for the first requirement.

With $\text{Wel}_{i'}(x', p') - \lambda_{i'} \cdot \text{Rev}_{i'}(x', p') > \text{Wel}_{i'}(x, p) - \lambda_{i'} \cdot \text{Rev}_{i'}(x, p)$ and $\text{Rev}_{i'}(x', p') \leq \text{Wel}_{i'}(x', p')$, we know the second requirement for $b'_{i'}$ dominating $b_{i'}$ is satisfied. Thus, we have $b'_{i'}$ dominates $b_{i'}$. And we get a contradiction. $\square$

*Proof.* (of Lemma 4.7) We prove by contradiction. Suppose that there exists some $b \in \Theta$ such that $b_{i',j'} < r_{i',j'}$ for some $i' \in [n], j' \in [m]$. Define $b'_{i'}$ to be the same as $b_{i'}$ except $b'_{i',j'} = r_{i',j'}$. We want to show $b'_{i'}$ dominates $b_{i'}$.

We start with proving the first requirement for $b'_{i'}$ dominating $b_{i'}$. For any $b'_{-i'}$, let $x, p$ be the allocation of bids $(b_{i'}, b'_{-i'})$, and $x', p'$ be the allocation of bids $(b'_{i'}, b'_{-i'})$. Since bids are the same in auctions other than $j'$, we know that $x_{i',j,k} = x'_{i',j,k}$ and $p_{i',j} = p'_{i',j}$ for any $j \neq j'$ and $k \in [s_j]$. Since $b_{i',j'} < r_{i',j'}$, we know bidder $i'$ is not allocated in auction $j'$ in $x$. So $p_{i',j'} = 0$ and $x_{i',j',k} = 0, \forall k \in [s_{j'}]$. Therefore, the payment difference for bidder $i$ is

$$\sum_{j=1}^{m}(p'_{i',j} - p_{i',j}) = p'_{i',j'}.$$

and the welfare difference for bidder $i$ is

$$\sum_{j=1}^{m}\sum_{k=1}^{s_j}(x'_{i',j,k} - x_{i',j,k}) \cdot v_{i',j} \cdot \text{pos}_{j,k} = \sum_{k=1}^{s_{j'}} x'_{i',j',k} \cdot v_{i',j'} \cdot \text{pos}_{j',k}.$$

We want to show

$$0 \leq p'_{i',j'} \leq \sum_{k=1}^{s_{j'}} x'_{i',j',k} \cdot v_{i',j'} \cdot \text{pos}_{j',k}.$$

There are two cases. The first case is that bidder $i'$ does not get allocated in auction $j'$ in allocation $x'$. In this case, we simply have

$$0 = p'_{i',j'} = \sum_{k=1}^{s_{j'}} x'_{i',j',k} \cdot v_{i',j'} \cdot \text{pos}_{j',k}.$$

In the second case, bidder $i'$ gets allocated to some slot $q'$ in auction $j'$ in allocation $x'$. We can write $\sum_{k=1}^{s_{j'}} x'_{i',j',k} \cdot v_{i',j'} \cdot \text{pos}_{j',k}$ as $\text{pos}_{j',q'} \cdot v_{i',j'} \cdot \gamma$. Since in GSP with reserve, the payment in each auction for each bidder is at most the bid multiplied by the position normalizer and we have bid $b'_{i',j'} = r_{i',j'}$, we get

$$0 \leq p'_{i',j'} \leq \sum_{k=1}^{s_{j'}} x'_{i',j',k} \cdot r_{i',j'} \cdot \text{pos}_{j',k} \leq \sum_{k=1}^{s_{j'}} x'_{i',j',k} \cdot v_{i',j'} \cdot \text{pos}_{j',k}.$$

Now we have

$$0 \leq \sum_{j=1}^{m}(p'_{i',j} - p_{i',j}) \leq \sum_{j=1}^{m}\sum_{k=1}^{s_j}(x'_{i',j,k} - x_{i',j,k}) \cdot v_{i',j} \cdot \text{pos}_{j,k}.$$

The rest of the proof follows exactly as the proof for the first requirement in Lemma 4.1.

Now we proceed to prove the second requirement for $b'_{i'}$ dominating $b_{i'}$. Consider the simple $b'_{-i'}$ with all zeroes. Again, let $x, p$ be the allocation of bids $(b_{i'}, b'_{-i'})$, and $x', p'$ be the allocation of bids $(b'_{i'}, b'_{-i'})$. For auction $j'$, clearly bidder $i'$ gets allocated a slot in $x'$ but not in $x$ due to $b_{i',j'} < r_{i',j'}$. And in $(x', p')$, this slot price is the reserve multiplied by the position normalizer which is lower than the value multiplied by the position normalizer. For other auction $j \neq j'$, bidder $i'$ gets the same allocation and price in $(x, p)$ and $(x', p')$. Therefore, we know

$$\text{Wel}_{i'}(x', p') - \lambda_{i'} \cdot \text{Rev}_{i'}(x', p') > \text{Wel}_{i'}(x, p) - \lambda_{i'} \cdot \text{Rev}_{i'}(x, p).$$

Since bidders other than bidder $i'$ bid all zeroes and there are no boosts, bidder $i'$ payment can be bounded by reserves, i.e.

$$\text{Rev}_{i'}(x', p') \leq \sum_{j=1}^{m} p'_{i',j} = \sum_{j=1}^{m} \sum_{k=1}^{s_j} x'_{i',j,k} \cdot r_{i',j} \cdot \text{pos}_{j,k} \leq \sum_{j=1}^{m} \sum_{k=1}^{s_j} x'_{i',j,k} \cdot v_{i',j} \cdot \text{pos}_{j,k} = \text{Wel}_{i'}(x', p').$$

With $\text{Wel}_{i'}(x', p') - \lambda_{i'} \cdot \text{Rev}_{i'}(x', p') > \text{Wel}_{i'}(x, p) - \lambda_{i'} \cdot \text{Rev}_{i'}(x, p)$ and $\text{Rev}_{i'}(x', p') \leq \text{Wel}_{i'}(x', p')$, we know the second requirement for $b'_{i'}$ dominating $b_{i'}$ is satisfied.

We get $b'_{i'}$ dominates $b_{i'}$. And then we get a contradiction. $\square$

## D  Missing Proofs of Section 4.3

*Proof.* (of Lemma 4.9) Notice that in the proof Lemma 4.7, the only property we use about GSP with reserve is that the payment in each auction for each bidder is at most its bid multiplied by the position normalizer. This property also holds in FPA with reserve. Therefore, Lemma 4.9 simply follows from the proof of Lemma 4.7. $\square$

## E  Instances with Matching Approximation Lower Bounds

First of all, for revenue approximation ratio, it is tight even for a setting with a single bidder and a single auction. Assume the bidder's value is 1 and the signal is between $\gamma$ and 1. The seller cannot set a reserve larger than the signal; or otherwise when the signal is 1, the reserve would be larger than the bidder's value, leading to a 0-approximation. Therefore, when the signal is $\gamma$, the reserve would be at most $\gamma$, leading to a $\gamma$-approximation in revenue.

For welfare approximation ratios, we show three two-bidder examples for the settings with reserve only, boost only, and reserve & boost. In all examples, buyers' targets are 1 and there are two auctions. The values of bidder 2 are always 0 for auction 1 and 1 for auction 2. We use different valuations of bidder 1 in both auctions to establish the lower bounds. $\epsilon > 0$ represents an arbitrarily small number.

### E.1  Tight instance for reserve only auctions

| | Values in Auction 1 | Values in Auction 2 |
|---|---|---|
| Bidder 1 | $1/(1-\gamma)$ | $\epsilon$ |
| Bidder 2 | 0 | 1 |

In the above example, the reserve signal for bidder 1 in auction 1 is $s \in [\gamma/(1-\gamma), 1/(1-\gamma)]$. When the realized signal $s = \gamma/(1-\gamma)$, the reserve is at most $s$, and bidder 1 has incentive to win both auctions, paying at most $\gamma/(1-\gamma) + 1 = 1/(1-\gamma)$, receiving $1/(1-\gamma) + \epsilon$ total value. This leads to $(1/(1-\gamma) + \epsilon)/(1/(1-\gamma) + 1) = 1/(2-\gamma)$ in welfare approximation.

Moreover, as the signal could be as high as bidder 1's value in auction 1, the reserve multiplier cannot be larger than 1; or otherwise, bidder 1 would choose to skip the first auction. By using a reserve multiplier that is at most 1, there is a realization of the signal in which the reserve is at most $\gamma/(1-\gamma)$.

### E.2  Tight instance for boost only auctions

| | Values in Auction 1 | Values in Auction 2 |
|---|---|---|
| Bidder 1 | $1 - \gamma + \epsilon$ | $\gamma$ |
| Bidder 2 | 0 | 1 |

When there is no boost, bidder 1 has the incentive to win both auctions, paying 1 and receiving $(1 - \gamma + \epsilon) + \gamma = 1 + \epsilon$ total value. This leads to an approximation ratio $1/(2 - \gamma)$ in welfare.

In the worst case scenario, consider that the boost signals in auction 2 are $s = \gamma$ for both bidders; and thus, boosts are not effective in changing the auction outcome for auction 2.

### E.3   Tight instance for auctions with both reserves & boost

|          | Values in Auction 1 | Values in Auction 2 |
|----------|:-------------------:|:-------------------:|
| Bidder 1 | $1 + \epsilon$      | $\gamma$            |
| Bidder 2 | $0$                 | $1$                 |

When there is no boost and the reserve for bidder 1 in auction 1 is at most $\gamma$ (realized signal $s = \gamma$), bidder 1 has the incentive to win both auctions, paying at most $\gamma + 1$ and receiving $(1 + \epsilon) + \gamma$ total value. This leads to an approximation ratio $(1 + \gamma)/2$ in welfare. In the worst case scenario, consider that the boost signals in auction 2 are $s = \gamma$ for both bidders; and thus, boosts are not effective in changing the auction outcome for auction 2. As for the reserves, similar to our analysis in the first example, the seller needs to set a reserve that is at most $\gamma$ for bidder 1 in auction 1 for some instances.