# OpenReview forum: "Robust Auction Design in the Auto-bidding World"
_NeurIPS.cc/2021/Conference — NeurIPS 2021 Poster_

### Official Review · Reviewer_xr54 · 2021-07-14

**Rating:** 6
**Confidence:** 4

**Summary:**

This paper studies welfare and revenue outcomes in a position auction with boosting and personalized reserves, where bidders behave in a manner motivated by autobidding.  Specifically, each bidder is assumed to want to maximize a modified utility of the form (value - lambda * payment) where lambda lies in [0,1], subject to an ROI constraint that value is at least payment.  Lambda = 1 corresponds to standard quasi-linear utility while lambda = 0 corresponds to value maximization.  As a solution concept, the paper considers all profiles of undominated bids, which is a relaxation of equilibrium notions.  The underlying auction is either VCG or GSP.

Boosting and personalized reserves aim to improve efficiency and welfare by leveraging information about an agent's true value and incorporating it into the auction rule.  This paper asks what happens when this information is noisy, which they model as having the reserve price and boosting parameters be within some constant factor of the agent's true value.  The authors show that when this holds, the welfare and revenue will be approximately optimal under arbitrary undominated bids.

The paper concludes with an experimental study of the proposed method on synthetic data based on historical bidding traces.  The study indeed shows that natural bidding behavior converges with the addition of approximate reserves and boosts, and that welfare and revenue improve.


**Limitations And Societal Impact:**

In my opinion, boosting and personalized reviews should be viewed with wary skepticism, as they essentially bend incentive properties to gain welfare and revenue.  There is a sense in which platforms exert monopoly power when using these tools, since they typically push the auction toward a platform-favorable market outcome.  This paper mostly ignores the way in which the platform learns information about buyer values, and hence sidesteps these issues.  Some acknowledgement and/or discussion about this would be helpful.

**Main Review:**

Boosting and personalized reserves are common in modern advertising auctions, and evaluating the impact of noisy estimates in such systems is an interesting direction.  Prior work (e.g., Deng et al. 2021b) had shown that these approaches are consistent with non-quasi-linear objectives used in autobidding systems.  This work builds upon that line by showing that the conclusions extend to scenarios where different auction participants have different types (utility-maximizer vs value-maximizer), and to the broader space of undominated bid profiles.

The authors position the paper as focusing on the case with boosts and reserves coming from noisy signals, such as perhaps being learned imperfectly by the platform.  However, this is modeled as assuming a strict constraint on boost factors and reserve prices that relate them to a bidder's true value.  This doesn't feel like a perfect fit; I would have preferred to see the signals as part of the model, perhaps taken from a distribution that allows very incorrect estimates with low probability.

The paper also avoids discussing incentive issues that arise from the use of such factors.  This is especially relevant in an autobidding world, where advertisers might rightly be concerned that the platform exploits the autobidding interface to learn information that can be used against the bidder's interests.  I would have liked to see at least a passing discussion of such issues.

The welfare and revenue bounds themselves are not particularly surprising, and follow using now-standard logic analyzing the impact of boosting and reserves.  What is nice about the paper is the generality of the model in which these bounds hold.  The experimental evaluations are a nice addition as well, especially for exploring convergence properties (a notable gap in the theory).

Overall, I think this is a modest but helpful contribution to our understanding of the robustness of position auction tricks like boosting and personalized reserve tuning.  The modeling omits many important practical issues, but the theory presented here makes a reasonable start that could be fleshed out with further work.  The technical contribution itself shares much in common with recent prior work.  Overall I think the paper could be accepted to the conference if there is room.

===

Post Author Response: Thanks for the thoughtful response.  I agree that adding the citations and discussion mentioned would be helpful. Overall I'm still favorable about the paper.


**Time Spent Reviewing:**

2

---

> ### Author Response · Authors · 2021-08-10
> **Author Response to Reviewer xr54**
>
> **Regarding the contribution of the paper**:
> In addition to the noisy signals, the main focus of this paper is to understand the impact of reserves in the autobidding world while the prior work only studies boosts (e.g., Deng et al. 2021b). Note that boosts and reserves are fairly different components in auctions. Reserves filter out the candidates whose bids are lower than their reserves and then the remaining candidates are ranked according to their original bids. In contrast, boosts are added to the candidates’ ranking scores, and therefore, the candidates are ranked according to their original bids plus their boosts.
>
> **Regarding the noisy signal**:
>
> For automated bidders, their objectives are mostly maximizing the number of conversions or clicks (e.g. Google: https://support.google.com/google-ads/answer/2979071?hl=en, Facebook: https://www.facebook.com/business/m/one-sheeters/facebook-bid-strategy-guide). Here, the value per auction would be the conversion rate or click rate. For these values, the platform (i.e., seller) could possibly build a model to learn these values and their models could have better estimations of values than the advertisers. More motivations can be found in recent papers (e.g. [Aggarwal et al. WINE’2019; Deng et al. WWW’2021; Balseiro et al. EC’2021]). And we will add more details and cite those papers for further motivations.
>
> Even though there are all the above papers, we  understand the concerns about pure value-maximizing buyers. That is part of the reason that we also consider settings with a mixture of value-maximizing and utility-maximizing buyers and also buyers with a combination of value and quasilinear utility. It is indeed interesting that the results for reserve prices are robust against the types of buyers.
>
> **Regarding the incentive issue**:
> Incentive issue is very important when the seller uses advertiser-related signals in auctions.
> In this paper, we are agnostic about how the signals are learned. Although out of the scope of this paper, the line of work on incentive-aware learning [Epasto et al. WWW’2018; Golrezaei et al. OR’2021] could be a starting point on how one can build a model that avoids or mitigates the incentive issues here.

---

### Official Review · Reviewer_C57R · 2021-07-14

**Rating:** 7
**Confidence:** 4

**Summary:**

This paper studies auction design for advertising slots when the agents are using auto-bidding optimization algorithms.

The bidders: The focus is on “value-maximizing agents”: this type of agent aims to maximize the total value he gets from his allocation, subject to the constraint that his value is at least his price (as opposed to a utility-maximizing agent, who aims to maximize his value minus his price; the authors also study interpolations between value maximizers and utility maximizers). The authors write that this is the prevalent adopted model for the behavior of auto-bidding agents. There are $m$ position auctions and each agent $i$ has a fixed (i.e., not Bayesian) value $v_{i,j}$ for each auction $j \in [m]$. Agent $i$’s value for the $k^{th}$ slot in auction $j$ is $v_{i,j}\cdot pos_{j,k}$, where $pos_{j,k}$ is a position normalizer that doesn’t depend on $i$ and is common knowledge.

The auctions: The paper studies VCG, GSP, and first-price auctions with reserves and boosts, the latter of which means that each bidder’s bid $b_{i,j}$ is increased by some boost $z_{i,j}$. The reserves and boosts are set to be equal to some noisy signals about the values $v_{i,j}$ which the authors write could be obtained, for example, using a machine learning model. These signals are assumed to be accurate up to some multiplicative error. Also, the bids are assumed to be undominated (which includes the support of Nash equilibrium bids as a subset).

The results: The authors prove approximation guarantees showing that for each of these three auctions, the revenue and welfare are approximately optimal, where the approximation factor depends on the accuracy of the noisy value signals. They also provide experiments on semi-synthetic data that validate these theoretical findings. From my understanding, the buyers’ values are based on real-world ad auction data, but the signals and bids are synthetically generated.

**Limitations And Societal Impact:**

I appreciate the limitations the authors describe in the final section---namely, that to set the reserve prices, it is necessary to know how noisy the signals are.

**Main Review:**

To start, I appreciate the relaxations this paper has made with respect to prior research, as summarized in lines 39-47. In particular, I like that the results require only noisy knowledge of the buyers’ values, which seems plausible. I also appreciate that the results hold for a mixture of value maximizers and utility maximizers, rather than one or the other.

I did find some of the results a bit hard to interpret—in particular, Lemma 3.1, which is the first actual result of the paper. The lemma depends on quite a lot of conditions and parameters $\alpha, \beta, \mu, \eta$, so it was hard for me to wrap my head around if they were reasonable conditions. It would be helpful to remind the reader which conditions hold by definition of the model (e.g., the first condition, from my understanding) and which you will prove hold (e.g., the third condition).

There were also some aspects of the model I think could have used more explanation. For example, the authors write that it is common to assume the buyers are value maximizing in prior research on auto-bidding, but intuitively, why is this the case? Intuitively, why aren’t they utility maximizers?

Other than these two confusions, I thought the paper was pretty well written, though I also have some more detailed comments:
-	Lines 45-46: At first I was confused, thinking the signals were the only input to the auction. But it’s really signals together with actual bids, from my understanding.
-	Line 53: “As longs as” -> “as long as”
-	Line 114: “approximates” -> “approximate”
-	Line 145: I think this should be “wins slot k in auction j”
-	Line 154: This should be “the prices of”
-	Line 158: Why isn’t this $\hat{b}$? The other prices are in terms of $\hat{b}$ rather than $b$.
-	Line 165: “an one” -> “a one”
-	Lines 184-197: The notations x, p, x’, and p’ are used repeatedly in lines 185-195 but only defined in lines 196-197—it would be good to move this definition earlier.
-	Lines 306-307: What does “the dataset” consist of, the $v_{i,j}$s? Can you say a bit more about how it’s generated?
-	Line 307: $\delta$ is used in line 201 when defining uniform bidding strategies, but here $\alpha$ is used, so it would be good to make this notation consistent.
-	Lines 314-315: What are “spend” and “target spend” in this inequality? Do you mean Wel and Rev?
-	Line 318: “overtime” -> “over time”
-	Line 321: I think you mean “experiment with boosts”
-	Line 342: Could you mean welfare instead of revenue? I don’t see anything about revenue in Figure 1a.
-	Line 374: “requirement” -> “require”

**Time Spent Reviewing:**

4

---

> ### Author Response · Authors · 2021-08-10
> **Author Response to Reviewer C57R**
>
> Thanks for your detailed comments.
>
> We will make proofs easier to digest by reminding readers about the conditions we use.
>
> We will add more context and discussions about why prior work and we use value maximizers as a model for autobidders, for example, references to automated bidding products description page on major online advertising platforms (Google: https://support.google.com/google-ads/answer/2979071?hl=en, Facebook: https://www.facebook.com/business/m/one-sheeters/facebook-bid-strategy-guide).
>
> We will fix typos mentioned in the comments. In particular, regarding line 158, it’s indeed a typo and the price for FPA should be $(\hat b_{k,j} - z_{i,j})\cdot pos_{j,k}$ which is also equal to $b_{i,j} \cdot pos_{j,k}$.

---

### Official Review · Reviewer_m9PK · 2021-07-16

**Rating:** 5
**Confidence:** 4

**Summary:**

This paper studies the auction design when the buyers are either utility or value maximizer with budget constraint. Under this setting the authors focus on three mechanisms VCG, GSP and First Price Auction. They have two main contributions compared to the previous literature:
1. They show that introducing reserve prices might have a positive impact on social welfare.
2. Also the authors try to model that the seller does not know the exact values of the bidders, meaning has only access to noisy signals.

Then the authors assess their findings by running some simulations and running real experiments that confirm their findings.


**Main Review:**

I like the paper and it is interesting. However, I have some conceptual questions related to the significance of the contribution compared to the previous literature.
1. In Deng et al 2021, the authors have shown that boosting can make the auction more efficient in a similar setting. The present paper proposes reserve prices. By setting the boosting factor such that it matches the reserve price, especially in GPS and VCG, do the results of the boosting from the previous literature extend the results in the case when the seller knows the accurate values?
2.Regarding the noisy signal, the authors indeed show that efficiency extends modulo some error. This is important since indeed in practice it does not hold the fact that the seller knows the exact signals. However I do find this result surprising, the paper would improve significantly if the authors show that these bounds are tight and robust to any uncertainty set and not only a multiplicative one that implicitly still assumes that the seller knows the bounds of the inaccuracy that depend on the try signal.

A minor comment: it is great to have the table 1 summary, however it uses a lot of notation that is hard to grasp without the model. I would recommend either to move it after the model introduction or introduce with more details the notation.



**Time Spent Reviewing:**

8

---

> ### Author Response · Authors · 2021-08-10
> **Author Response to Reviewer m9PK**
>
> **Comparison with prior works**:
> Deng et al. [2021] study boosts with/without budget constraints. In this paper, we study mainly reserve prices based on noisy signals and have the settings with/without boost as one step further. Furthermore, we don’t study the setting with budget constraints. We also consider settings with a mixture of value-maximizing and utility-maximizing buyers.
>
> **Regarding whether the results for reserve prices are implied by previous work on boosts when there is no uncertainty in the signals**:
> Boosts and reserves are fairly different components in auctions. Reserves filter out the candidates whose bids are lower than their reserves and then the remaining candidates are ranked according to their original bids. In contrast, boosts are added to the candidates’ ranking scores, and therefore, the candidates are ranked according to their original bids plus their boosts. Thus, even when the signals are perfectly accurate, the results of boosts cannot be directly extended to reserves. The nature of the results are also different. For example, our results for reserve prices apply to a mixture of value-maximizing and utility-maximizing buyers, but some of our results related to boost apply only to value-maximizing buyers but do not apply to utility-maximizing buyers.
>
> **Regarding Table 1**:
> Thanks for the suggestions. We will make Table 1 easier to understand.
>
> **Regarding lower bounds**:
> We can prove lower bounds to show our analyses are tight and we will add them to the revised version of the paper. In particular, except the welfare approximation ratio for GSP with reserve, all the approximation ratios provided in Table 1 are tight.

---

> > ### Comment · Reviewer_m9PK · 2021-08-23
> > **Response**
> >
> > Thank you for your responses.
> >
> > Related to the question of comparison to previous works; the auction design/autobidding/budgeted bidding problem is studied by a number of industrial research labs in addition to the cited papers in the submission. For example, on auctions with budgets the authors might consider including
> >
> > - Contizer et al 2018. https://arxiv.org/abs/1706.07151
> > - Contizer et al 2019. https://arxiv.org/abs/1811.07166
> > - Avadhanula et al. 2021. https://arxiv.org/abs/2103.10246
> > - Celli et al 2021. https://arxiv.org/abs/2106.09503
> >
> > and references therein (+ follow-ups thereof).

---

> > > ### Author Response · Authors · 2021-08-23
> > > **Thanks**
> > >
> > > Thanks for pointing us to these related works. We will cite and discuss them in the revision.

---

### Official Review · Reviewer_7XKj · 2021-07-17

**Rating:** 6
**Confidence:** 3

**Summary:**

This paper focuses on value maximizing bidders with return on spend constraints—a paradigm that has drawn considerable attention recently as more advertisers adopt auto-bidding algorithms in advertising platforms, and show that the introduction of reserve prices has a novel impact on the market. Choosing reserve prices appropriately can improve not only the total revenue but also the total welfare.
Their results also demonstrate that reserve prices are robust to bidder types, i.e., reserve prices work well for different bidder types, such as value maximizers and utility maximizers, without using bidder type information. Then generalize these results for a variety of auction mechanisms such as VCG, GSP, and first-price auctions. Moreover, they show how to combine these results with additive boosts to improve the welfare of the outcomes of the auction further. Theoretical observations are complemented with an empirical study using data from online advertising auctions.


**Limitations And Societal Impact:**

Yes

**Main Review:**

Task is new, and the work is a novel combination of well-known techniques. The related works are adequately cited, the comparison is clear..
The submission is technically sound. Methods are used appropriately and it’s a complete work. The authors are careful and honest about evaluating both the strengths and weaknesses of their work.
Line 314, how this eta is chosen? Line 326 to 327, is there a redundant?
The submission is clearly written and well organized. It adequately inform the reader.
The results are important and advance the state of art.


**Time Spent Reviewing:**

5

---

> ### Author Response · Authors · 2021-08-10
> **Author Response to Reviewer 7XKj**
>
> Thanks for your insightful comments!
>
> **Regarding the learning rate $\eta$ in line 314**:
> We use geometrically decaying learning rates.
>
> **Regarding Line 326 to 327**:
> It describes the setup for the arm with both boosts and reserves. Intuitively, it is an arm that combines reserve-$\gamma$ and boost-$\gamma$.

---

### Decision · Program_Chairs · 2021-09-27

**Decision:**

Accept (Poster)

**Comment:**

All the reviewers agreed that this paper studies a worthwhile topic, and are generally positively disposed towards the paper. At the same time, several reviewers felt that the technical contributions of the paper are somewhat marginal.

On a relatively minor note, I will also echo what one reviewer said about related work: there is significant work by several industry labs on autobidding, and especially budget constraints. The present paper seems somewhat heavily skewed towards citing work primarily from Google. While this is probably due to an accident of only hearing about those particular works, it does seem like something the authors should take care to fix, as it looks a bit odd.

A specific early work that I want to point out is the paper by Borgs et al. "Dynamics of bid optimization in online advertisement auctions." Proceedings of the 16th international conference on World Wide Web. 2007.